# Distinguishing Plasmin-Generating Microvesicles: Tiny Messengers Involved in Fibrinolysis and Proteolysis

**DOI:** 10.3390/ijms24021571

**Published:** 2023-01-13

**Authors:** Laurent Plawinski, Audrey Cras, José Rubicel Hernández Lopez, Aurora de la Peña, Angéline Van der Heyden, Catherine Belle, Florence Toti, Eduardo Anglés-Cano

**Affiliations:** 1UMR 5797 Laboratoire de Physique des deux infinis, Université de Bordeaux-CNRS, 33170 Bordeaux, France; 2Assistance Publique-Hôpitaux de Paris, Hôpital Saint-Louis, Unité de Thérapie Cellulaire, 75610 Paris, France; 3Departamento de Farmacologia, Facultad de Medicina & Departamento de Biologia Molécular, Universidad National Autonoma de Mexico, Instituto Nacional de Cardiologia, Mexico City 04510, Mexico; 4Department of Molecular Chemistry (DMC), Université Grenoble-Alpes, CNRS, UMR 5250, 38000 Grenoble, France; 5Université de Strasbourg-INSERM, UMR 1260, Centre de Recherche en Biomédecine (CRBS), 67084 Strasbourg, France; 6Université Paris Cité, INSERM, Innovative Therapies in Haemostasis, 75013 Paris, France

**Keywords:** plasminogen, tPA, uPA, extracellular vesicles, microvesicles, zinc complexes, fibrinolysis crosstalk, pericellular proteolysis

## Abstract

A number of stressors and inflammatory mediators (cytokines, proteases, oxidative stress mediators) released during inflammation or ischemia stimulate and activate cells in blood, the vessel wall or tissues. The most well-known functional and phenotypic responses of activated cells are (1) the immediate expression and/or release of stored or newly synthesized bioactive molecules, and (2) membrane blebbing followed by release of microvesicles. An ultimate response, namely the formation of extracellular traps by neutrophils (NETs), is outside the scope of this work. The main objective of this article is to provide an overview on the mechanism of plasminogen reception and activation at the surface of cell-derived microvesicles, new actors in fibrinolysis and proteolysis. The role of microvesicle-bound plasmin in pathological settings involving inflammation, atherosclerosis, angiogenesis, and tumour growth, remains to be investigated. Further studies are necessary to determine if profibrinolytic microvesicles are involved in a finely regulated equilibrium with pro-coagulant microvesicles, which ensures a balanced haemostasis, leading to the maintenance of vascular patency.

## 1. Cell Activation: A Scenario for Fibrinolytic and/or Proteolytic Development

A number of stressors and inflammatory mediators (cytokines, proteases, oxidative stress mediators) released during inflammation or ischemia stimulate and activate cells in blood, the vessel wall, or tissues. The most well-known functional and phenotypic responses of activated cells are (1) the immediate expression and/or release of stored or newly synthesized bioactive molecules, and (2) membrane blebbing followed by the release of microvesicles. An ultimate response, namely the formation of extracellular traps by neutrophils (NETs), is outside the scope of this work [1].

For instance, activated cells express membrane proteases that produce pericellular proteolytic activity or are able to excise mature growth factors from its plasma membrane-bound precursors [2]. Of particular interest within this context is the response of cells that express plasminogen activators, either tPA (tissue plasminogen activator) or uPA (urokinase plasminogen activator) also known as scuPA (single chain uPA). Current knowledge indicates that tPA is primarily involved in fibrinolysis, i.e., the activation of plasminogen at the fibrin surface, while uPA is colocalised with plasminogen at the cell membrane, resulting in the generation of plasmin and pericellular proteolysis [3]. Accordingly, finely regulated plasmin formation by uPA on cells plays an essential role in extracellular matrix (ECM) remodelling, cell migration, and survival [4]. Notwithstanding, receptor-bound plasminogen may also be transformed into plasmin on cells that express tPA [5] whereas uPA may exhibit fibrinolytic activity via activation of plasminogen bound to carboxy-terminal lysine (C-ter-Lys) residues on partially degraded fibrin [6]. Furthermore, plasminogen-independent targets for both tPA and uPA have been proposed in a number of in vitro and in vivo studies (uPA: platelet derived growth factor-D, PDGF-D; tPA: PDGF-C, and the N-methyl-D-aspartate receptor, NMDAR) [7,8,9]. 

## 2. Cell Activation: Membrane Blebbing and Microvesicle Release

Beyond the expression and release of bioactive molecules, another early manifestation of cell activation is the structural and local modification of the membrane. Well-identified changes span from externalization of phosphatidylserine and phosphatidylethanolamine, changes in lipid leaflet composition, alteration of the interactions with the cytoskeleton and clustering of lipids and transmembrane proteins. These lipid changes in the plasma membrane lead to local bending and direct outward budding of the membrane. The emerging membrane bud ultimately results in the release of membrane microvesicles into the extracellular space. (Figure 1). Exposure of phosphatidylserine at the outer leaflet of the membrane is induced by an increase in calcium concentrations, which is also required for the activation of calpains [10,11]. The calcium cytoplasmic raise promotes a disordered state in the concerted activities of phospholipid membrane transporters (floppases, flippases, and scramblase) that maintain the membrane phospholipid asymmetry of resting cells [12,13,14]. The ATP-dependent flippases (e.g., aminophospholipid translocase, P4 ATPase) and floppases (including the ATP-binding cassette transporter C1, ABCC1) are respectively inward- and outward-directed transporters, whereas the calcium-dependent scramblase TMEM16F (transmembrane protein 16F,) [15] or the caspase-dependent XKR8 scramblase [16] facilitate bidirectional movement between the membrane leaflets. In a disordered state, phosphatidylserine and phosphatiylethanolamine are rapidly translocated from the inner to the outer membrane leaflet, leading to a randomized distribution of aminophospholipids between the two layers. Compared to the phosphatidylserine and phosphatidylethanolamine externalization, the reverse transport of phosphatidylcholine is slow and unable to contain the phospholipid transient overload of the outer leaflet. Occurring within the plane of the membrane, the formation of lipid rafts is another contributive mechanism to microvesicle emission. While providing a platform for the optimal assembly of phospholipid transmembrane transporters and their regulatory elements [17,18,19], they also enable cell membrane curvature [16,17,18,19,20,21]. Furthermore, intracellular calcium stimulates the activity of calpains that cleave the actin filaments, thereby limiting the retention strength of the cytoskeleton, which will no longer counterbalance the phospholipid overload of the outer leaflet [22,23]. In parallel with the weakening of the cytoskeleton by calpains, there is actin-myosin-based contraction induced by the ARF-1/RhoA/ROCK/MLCK cascade pathways, which thereby facilitate membrane blebbing and shedding of the membrane fragments-bound bags (Figure 1) [24,25]. 

Such membrane bags form extracellular vesicles of less than 1 μm and above 100 nm, hereafter named microvesicles. In addition to phosphatidylserine, microvesicles carry identity glycoproteins and bioactive molecules: RNA or proteins such as tissue factor (TF) [26,27]. Thus, in the vascular territory, phosphatidylserine exposed on microvesicles serve as a functional catalytic surface for the assembly and activation of blood coagulation factor complexes, thus further promoting in situ haemostasis, a physiological function of activated platelets and shed platelet-derived microvesicles [28]. 

Some studies have shown that besides their well-known procoagulant activity, cell-derived microvesicles carry plasminogen activators (either tPA or uPA) as well as a receptor for uPA (uPAR) and receptors for plasminogen [29,30]. The potential for plasminogen activation on microvesicles is of particular interest as they may convey this activity and plasmin to distant sites. Microvesicles may thereby induce extracellular proteolysis and fibrinolysis, intermediary mechanisms in angiogenesis and thrombus lysis. 

One of the objectives of this article is to provide an overview on (i) the consequences of cellular plasmin formation on cell adherence and survival and (ii) the mechanism of plasminogen reception and activation at the surface of cell-derived microvesicles, new actors in fibrinolysis and proteolysis. 

## 3. Receptor-Bound Plasmin Induces Cell Detachment and Apoptosis 

ECM macromolecules such as fibronectin, laminin, collagen IV, and tenascin act as a scaffold surface for focal adhesions via coupling with integral membrane protein receptors linked to the intracellular cytoskeleton [31]. In addition, integrins transmit outside-in signalling, which determines cell growth, morphology, motility, and anchorage-dependent survival. Disruption of the ECM-integrin interactions by proteolysis may have consequences on cell adhesion and survival [32]. A number of recent studies have clearly demonstrated that plasmin formation on cells induces cell detachment and apoptosis [33,34,35,36,37,38,39,40,41,42]. Indeed, stimulated adherent or migrating cells that synthesize plasminogen activators also express plasminogen receptors (e.g., α-enolase, annexin A2-S100A10, histone H2B, Plg-R_KT_) and develop the capacity to generate plasmin at their surface upon incubation with plasminogen [43,44,45,46,47,48]. Plasminogen receptors have in common exposed carboxy-terminal lysine residues that interact with the lysine-binding site of plasminogen kringle 1 [49]. Therefore, lysine analogues such as ε-aminocaproic acid or tranexamic acid (TXA) block the LBS of plasminogen kringle 1 and specifically inhibit binding of plasminogen to its receptor and thereby plasmin formation (Figure 2B, inset) [33,34,38,41]. Kinetic studies on plasmin formation (Figure 2A) have shown that plasminogen incubated with cells is assembled at the membrane and transformed into plasmin in a time-, lysine-, and dose- dependent manner until saturation (Figure 2B). 

These results are comparable to those obtained with different cell lines including primary endothelial cells, myofibroblasts, monocytes, or various cell lines [41,51,52,53].

While the Lysine-analogues block plasminogen kringle 1, the carboxypeptidase B was recognized to cleave C-ter-Lysine residues of several plasminogen receptors [42]. It has been suggested that this effect may be mediated by an unstable carboxypeptidase present in human plasma [54]. Additional modulation of the plasminogen binding to distinct receptors can be achieved with the use of specific antibodies, e.g., antibodies directed against α-enolase [43]. Altogether, pericellular proteolysis appears to be dependent on cell membrane-generated plasmin. Pericellular proteolysis associated to cell membrane plasmin is therefore a surface-controlled process that plays an essential role in ECM remodelling, cell migration, and survival [55,56]. 

The survival of cells within structural-functional units involving tissue specific components and the microvasculature (e.g., neurovascular unit, glomerulus, pulmonary alveolus, islet of Langerhans, and liver lobules) depends on dynamic cell–matrix interactions that ensure their adhesion to the substratum and tissue cohesion [42]. For instance, in the absence of any ECM interaction, human endothelial cells and myofibroblasts rapidly enter apoptosis [32,41]. Disruption of the balance between plasminogen activators and their inhibitors could be a trigger for increased cellular generation of plasmin and the modulation of the fate of cells at the vicinity. Accordingly, excessive proteolysis of the ECM by cells that express a plasminogen activator system results, after prolonged exposure to plasminogen, in the loss of cell anchorage and subsequent apoptosis (Figure 3A) [33,34,35,36,37,38,39,40,41,42,49]. 

Cell retraction and detachment from the substratum is secondary to degradation of ECM proteins such as fibronectin and laminin, which participate in cell anchorage and survival signalling (activation of FAK, focal adhesion kinase, expression of anti-apoptotic genes such as bcl-2) [57]. Degradation of ECM components by plasmin leads to disruption of survival signals and eventually triggers programmed cell death characterized by a shortened cell survival, DNA fragmentation, caspase activity, and typical cell apoptotic features (Figure 3A) [42]. Electron microscopy allows for a detailed observation of cell morphology (Figure 3B) and characteristic apoptotic changes (Figure 3C). In growing cells (not shown), the nucleus membrane is clearly delimitated and the characteristics of normal cytoplasm are maintained. In plasminogen-treated cells (Figure 3C), the nucleus shows chromatin condensation, the cytoplasm is disorganized, contains lysis vesicles, and mitochondria become electron dense (Figure 3C, bottom micrographs). With prolonged stimulation, the chromatin shows a higher condensation and fragmentation (Figure 3C, upper micrographs). 

This plasminogen activation-dependent sequence (matrix degradation, cell detachment and retraction, apoptosis) has been observed in a variety of cellular types in vitro [33,34,35,37,39,40,41,58,59] and in animal models in vivo [36,60]. It could therefore be of relevance in pathological situations such as atherosclerosis and abdominal aortic aneurysm [61,62]. This sequence has also been implicated in mouse myocardial tissue degradation and left ventricular remodelling leading to cardiac insufficiency [63].

Specific cell types such as neurons and myofibroblasts display a distinct response to plasmin formation [41,50]. In contrast to adherent cells that are fully dissociated by plasmin (Figure 3A) [33,34,35,39], cortical neurons evolve to form multicellular aggregates (cell clusters) interconnected by fasciculating fibres, resulting in retraction of the monolayer that detaches from the matrix as a whole and single body (Figure 3B). Neurons in these cell clusters survive longer than isolated detached cells. Interestingly, plasmin-induced neuron detachment did not affect phosphorylation of FAK, thus suggesting that cell-to-cell adhesion could participate in the stimulation of FAK phosphorylation and thereby decrease their susceptibility to cell detachment-induced apoptosis [50].

Proteolysis of ECM components and cell detachment-induced apoptosis can be efficiently prevented in the presence of the serine protease inhibitors PN-1, PAI-1 and α_2_-antiplasmin or by inhibiting plasminogen binding to receptors with ε-ACA or TXA [34,35,36,37,38,49,59]. Accordingly, induction of PAI-1 secretion by TGF-β1 has been shown to reduce plasminogen-dependent apoptosis of fibroblasts and to promote myofibroblasts survival in chronic fibrotic disorders [40]. Indeed, a restriction in plasmin formation due to inhibition of uPA by an excess of PAI-1 worsens fibrosis [64]. This mechanism has been proposed for the observed fibrogenesis in lungs, kidneys, the heart, and the liver [64,65,66,67,68]. TGF-β1 and PAI-1 has been indeed qualified as potent fibrosis-promoting glycoproteins. Expression and activation of TGF-β leads to overexpression of PAI-1 via intermediary reactive oxygen species [65]. The fibrogenic effects of PAI-1 are related (i) to its anti-protease activity (uPA, tPA), (ii) to interactions with uPAR and its integrin co-receptors, leading to recruitment of interstitial macrophages and ECM producing myofibroblasts and (iii) to the inhibition of release and activation of anti-fibrotic hepatocyte growth factor [65,67].

## 4. Receptor-Bound Plasmin Induces Membrane Blebbing and Release of Microvesicles 

Besides the proteolytic activity of plasmin on ECM components, the first and immediate consequence of in situ plasmin formation is membrane blebbing [42]. This short-lived phenomenon has been observed at the initial phase of plasminogen incubation with mouse cortical neurons and was visualized by electron microscopy (Figure 4, upper micrographs). Blebbing of the membrane is followed by the release of microvesicles (Figure 4, bottom micrographs). 

Only a reduced number of microvesicles is detected in the absence of plasminogen. However, after incubation with plasminogen, the number of microvesicles increases dramatically in a concentration- and time-dependent manner. Visualised by electron microscopy, these microvesicles are around 300 nm in size and contain electron dense substructures surrounded by a well-defined membrane (Figure 4, bottom right). This vesiculation is prevented in the presence of aprotinin or lysine-analogues, thus indicating that membrane blebbing is a response to receptor-bound plasmin. Of note, the released microvesicles carry the plasminogen activator (tPA or uPA) of the parental cell and receptor-bound plasmin [42]. 

## 5. Beyond Coagulation, Microvesicles Are Dynamic Fibrinolytic Vectors 

Most clinical studies have focused on the procoagulant role of cell-derived microvesicles as a determinant of the risk of cardio- and cerebrovascular ischemic accidents and other thrombotic associated disorders [69,70,71,72]. Since microvesicles also convey other bioactive molecules (growth factors, receptors, inflammatory mediators), they are currently considered as a storage pool of bioactive effectors [73]. Indeed, it has recently been suggested that endothelial-derived microvesicles may also express anticoagulant or profibrinolytic properties, thereby complementing their procoagulant activity. The anticoagulant property of microvesicles is based, in part, on their ability to promote activation of protein C by thrombin [74,75]. Endothelial microvesicles also harbour matrix metalloproteinases [76] involved in extracellular matrix degradation, leading to disruption of the blood brain barrier integrity and ultimately to potential inflammation of the central nervous system (recently reviewed by Gassama Y and Favereaux A, 2021) [77] Microvesicles from the atherosclerotic plaque bear the TNF-α-converting enzyme (TACE) that is able to enhance endothelial cell surface processing of TNF-α and endothelial protein C receptor [78]. The recent discovery of a profibrinolytic activity on microvesicles adds further to their contribution in the maintenance of vascular integrity. Microvesicles shed by TNFα-stimulated human microvascular endothelial cells (HMEC-1), serve indeed as a surface for the assembly of plasminogen and its conversion into plasmin by uPA bound to its receptor (uPAR) (Figure 5A) [29]. 

The kinetics of plasminogen activation on these microvesicles is similar to the kinetics of plasmin formation on cells expressing uPA and uPAR [79]. The presence of uPA (and not tPA) on these endothelial microvesicles is due to the atypical synthesis of this plasminogen activator by the modified cell line HMEC-1, which was used as a model to generate microvesicles after TNFα stimulation [29]. The capacity of endothelial microvesicles to promote plasmin generation confers them new profibrinolytic and, in concert with matrix metalloproteinases, proteolytic functions [80] of relevance in fibrinolysis, cell migration, angiogenesis, and dissemination of malignant cells. 

Microvesicles were also shown to bind exogenous scuPA, thus suggesting that besides receptors available for plasminogen binding, the microvesicle surface also bears unoccupied uPAR molecules [29]. These data suggest that plasmin generation by microvesicles can be amplified by uPA transferred from the local environment to the surface of microvesicles exposing uPAR. This mechanism may have implications in tumour angiogenesis where uPA secretion is promoted. Thus, microvesicle from tumour cells could participate in the amplification of proteolytic processes at play in tumour growth [81].

Binding of plasminogen to the surface of microvesicles is mediated by C-ter-Lys residues as indicated by inhibition experiments using the lysine analogue ε-ACA, carboxypeptidase B and a monoclonal antibody directed against α-enolase (see above) (Figure 5B). The lysine-dependent binding of plasminogen was confirmed by the inhibitory effect of ε-ACA, while C-terminal lysine residues-dependent inhibition by carboxypeptidase B indicated that plasminogen activation was dependent on cell surface binding. This mechanism and the nature of the receptor were further confirmed by inhibition of plasmin generation with a monoclonal antibody directed against α-enolase, a major plasminogen binding protein on cell surfaces [29,82]. Recent data indicate that exposure of phosphatidylserine promotes localization of the plasminogen receptor histone 2B at the cellular membrane, suggesting that this receptor may also be found on microvesicles [83].

## 6. Microvesicles Bearing uPA Induce a Fibrinolytic Cross-Talk

Conformational transition of the plasminogen from the circulating compact closed conformation to the open one is key to the mechanism of its transformation into plasmin by plasminogen activators. The transition occurs in the solution when the lysine-binding sites of plasminogen kringles are saturated with ε-ACA or TXA. The induction of an open conformation by this lysine analogue has previously been shown and is well described in the literature [84,85]. More importantly, this transition operates when the plasminogen is directly bound to C-terminal lysine residues on fibrin or on its cellular receptors [84,86,87]. Plasminogen binding to its receptors is indeed a prerequisite for its efficient transformation into plasmin by a plasminogen activator localized on the same host surface, either fibrin or the cell membrane. In situ molecular co-assembly of plasminogen and its activators on cell receptors or on binding sites present on macromolecular complexes (fibrin or matrix surfaces) [88,89] is therefore key to fibrinolytic and pericellular proteolytic functions of plasmin.

Several lines of evidence indicate that uPA activates ε-ACA-liganded plasminogen faster than native plasminogen [90,91,92,93,94]. Furthermore, plasminogen bound to C-ter-Lys residues of fibrin is specifically recognized and activated by soluble scuPA [6], suggesting a relationship between its molecular conformation and plasmin generation. In a similar fashion, ε-ACA-liganded plasminogen (unable to bind to its receptor) is transformed into plasmin by cell-borne uPA/uPAR. Moreover, a mAb directed against the LBS of plasminogen kringle 1 that completely inhibits plasminogen binding and activation by tPA, does not prevent the formation of plasmin by cells bearing uPA [49]. To explain these particularities of plasmin formation by uPA, the existence of a new mechanism was recently proposed [95]. In this mechanism, the conformational transition of plasminogen bound to its receptor is essential for its activation by uPA. The recent description of the crystal structure of the compact form of native plasminogen is critical to understand why a transition to the open form is necessary for its transformation into plasmin [96]. This hypothesis was tested using fibrin, ECM glycoproteins, or Matrigel and platelets as support for plasminogen and uPA-bearing cell-derived microvesicles.

*Cross-talk on fibrin or ECM proteins.* Plasminogen bound to fibrin surfaces (Figure 6A), to extracellular matrix proteins (fibronectin, laminin) or to fibrin/fibronectin complexes (Figure 6B) is selectively recognized and activated into plasmin by uPA expressed on cells or carried by microvesicles. 

This mechanism of crosstalk may be of physiological relevance as it has recently been reported that monocytes may be involved in clot dissolution [97,98]. Since activated monocytes and macrophages release microvesicles bearing uPA, it is possible that these microvesicles will participate in activation of fibrin- or platelet-bound plasminogen. Indeed, leukocyte-derived microvesicles have been found in atherosclerotic plaques [99], where they can initiate fibrinolytic or proteolytic activities that may destabilize the atheroma plaque. A similar interaction may take place during inflammatory processes where primed cells could initiate a proteolytic cross-talk with plasminogen bound to other cells or to the matrix as suggested in the proposed model (see Figure 7B). The proposed crosstalk mechanism clearly explains the activation of fibrin-bound plasminogen by leukocytes reported recently [100].

*Cross-talk on platelet bound plasminogen.* Human microvesicle- or monocyte-borne uPA, but not tPA-bearing cells (neurons), were able to specifically activate platelet-bound Glu-plasminogen in a dose- and saturation-dependent manner (Figure 6C). The rate of plasmin formation on platelets by monocytes was two-fold higher than the activation of plasminogen bound to monocytes. This fibrinolytic cross-talk mechanism bypasses the requirement for assembly of profibrinolytic proteins on the same surface (Figure 7C), Ref. [95] introduces a complementary and new dimension for enhancement of fibrinolysis by platelets [51,102], and its efficiency suggests a potential physiological relevance. Thus, platelet-bound plasminogen activated by monocytes or microvesicles bearing uPA could be an additional source of plasmin in the fibrin clot, as suggested previously [97]. This mechanism is also in agreement with the recently reported platelet-dependent enhancement of lysis by scuPA, which identified platelet-bound plasminogen as the essential player [103].

*Cross-talk on Matrigel.* In co-culture experiments, fibrinolytic microvesicles affect endothelial progenitor cell angiogenesis in Matrigel via a fibrinolytic crosstalk [29]. Tube formation was stimulated at low concentrations of microvesicles whereas higher concentrations impaired the tube formation. Since plasminogen is present in the Matrigel [104], this dual effect may be related to crosstalk activation by microvesicles bearing uPA, as indicated by its prevention by an anti-uPA polyclonal antibody. The proangiogenic effect of fibrinolytic microvesicles is consistent with plasmin associated proteolytic activity, which favours cell migration via extracellular matrix processing. Other mechanisms may also be involved as plasmin may also affect angiogenesis indirectly through activation of matrix metalloproteinases [76]. High concentrations of microvesicles produce high amounts of plasmin and a dose-dependent decrease in tube formation by endothelial progenitor cells, which is in line with the ECM degradation, alteration of cell anchorage, and apoptosis caused by excessive plasmin generation [33,34]. Morphological changes in endothelial progenitor cells co-cultured with high amounts of microvesicles were indeed observed: accumulation of round and retracted cells evoke cell detachment, an effect that precedes apoptotic cell death [36]. 

Altogether these data indicate that Glu-plasminogen bound to C-ter Lys residues of platelets, fibrin, or ECM is recognized and transformed into plasmin by uPA anchored on monocytes or microvesicles. This mechanism bypasses the requirement for molecular co-assembly of plasminogen and its activator on the same surface, via a recognition and proteolytic cross-talk pathway [95]. Because plasmin is efficiently generated on platelets or on matrix surfaces by uPA-bearing cells or its microvesicles owing to crosstalk-mediated plasminogen activation, such a mechanism may be of potential physiologic relevance in fibrinolysis or proteolysis of ECM components (see the proposed model, Figure 7). In this case, plasminogen bound to its receptor in its open extended form is mandatory. However, the process essentially differs from classical activation of plasminogen on biological surfaces in that (i) the uPA is expressed on neighbouring cells or is carried by microvesicles and (ii) cells bearing tPA do not reproduce this effect.

## 7. Recent Developments 

*Proteases, microRNA, and microvesicles.* Besides the above-mentioned proteases and mediators, microvesicles may play an important role in the transport and regulation of microRNAs (miRNA) that regulate the translation of mRNAs and proteins [105,106]. These small intravesicular non-coding RNAs are thus protected from the action of plasma RNases and their breakdown is reduced. Different studies show the possibility of their involvement in thrombotic pathology or in fibrinolytic imbalance. For instance, microvesicles from neutrophils carry miR-155S, which favors NF-κB activation, contributing to vascular inflammation, atherogenesis, and atherosclerotic plaque formation [107].

*Detection of proteases on microvesicles.* Among current tests for microvesicle detection, biophysical approaches (such as flow cytometry, nano-particle Traking Analysis (NTA), Tunable Resistive Pulse Sensing (TRPS) can directly measure the size distribution and number of microvesicles. Flow cytometry has been largely used but the accuracy remains challenging for particles smaller than 300 nm and the presence of large protein complexes that overlap in biophysical properties (size, light scattering, and sedimentation) with microvesicles [108,109]. Functional assays such as the phosphatidylserine-capture annexin assay provide accurate data and high throughput capacities but have a number of limitations such as the Ca^2+^ dependence of the interaction and the sensitivity. Recently, we developed an assay where microvesicles are directly captured by a coordination complex immobilized onto a solid surface at physiological pH [110]. This approach requires the preferential recognition of coordination dinuclear zinc complexes by phosphates anions of phosphatidylserine exposed at the outer leaflet membrane microvesicles. The immobilization of such complexes or a related variant on a solid surface do not present complications encountered with the use of a biological molecule, such as interference with antibodies or protein complexes, calcium sensitivity, and even proteolysis encountered with the use of a biological molecule [111].

## 8. Concluding Remark 

Collectively, the above data indicate that microvesicles bind plasminogen and provide a catalytic surface for plasmin generation. Cell-derived microvesicles are thus identified as new actors in the plasminogen activation system. The role of microvesicle-bound plasmin in pathological settings involving inflammation, atherosclerosis, angiogenesis and tumour growth, remains to be investigated. The high concentration of microvesicles reported in atherosclerotic plaques suggests that plasmin generation on macrovesicles could participate in the modulation of the cell apoptosis/angiogenesis balance, influencing the plaque vulnerability. Further studies are necessary to determine if profibrinolytic microvesicles are involved in a finely regulated equilibrium with pro-coagulant microvesicles, which ensures a balanced haemostasis, leading to the maintenance of vascular patency. 

## Figures and Tables

**Figure 1 ijms-24-01571-f001:**
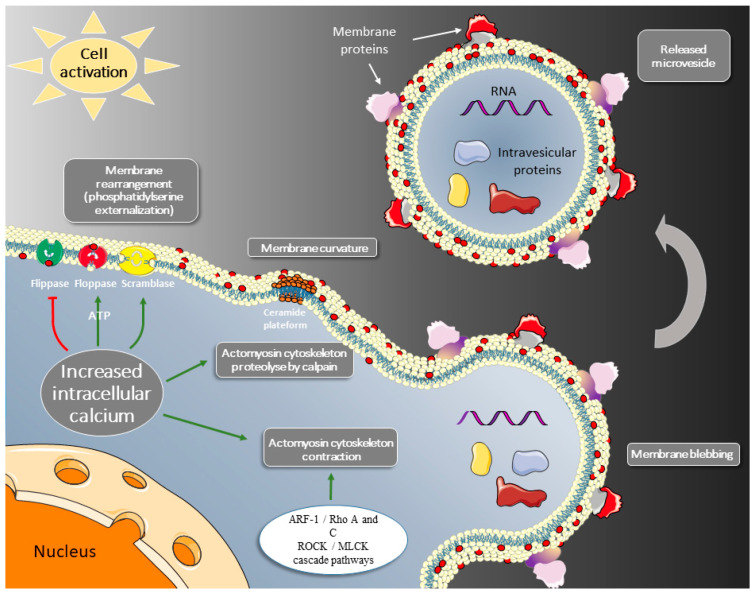
As a result of cell activation, the increase in intracellular calcium induces (i) exposure of phosphatidylserine at the outer leaflet of the membrane (red spot) through inhibition of flippase and scramblase-mediated transport, and ATP-dependent activation of floppase, (ii) membrane bending at the local ceramide-rich site, and (iii) contraction of the actomyosin cytoskeleton. Shed microvesicles constitute a catalytic surface for the assembly of coagulation factors and carry components of the parent cell. They distribute in the extracellular space and are thus able to induce, away from the parent cell, procoagulant (tissue factor), or fibrinolytic (plasminogen, uPA) activities.

**Figure 2 ijms-24-01571-f002:**
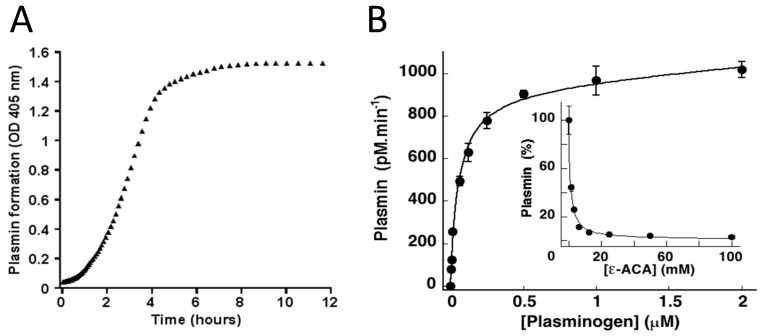
Plasminogen binding and activation on the membrane of mouse cortical neurons. (**A**) Kinetics of plasmin formation on the cell membrane upon incubation with a fixed amount of plasminogen. (**B**) Plasminogen binds to membrane receptors in a concentration-dependent manner and is converted to plasmin until saturation. The specificity of plasminogen binding to the terminal lysine residues of membrane receptors is confirmed by its inhibition in the presence of the lysine analogue ε-aminocaproic acid, ε-ACA (inset) [42,49,50].

**Figure 3 ijms-24-01571-f003:**
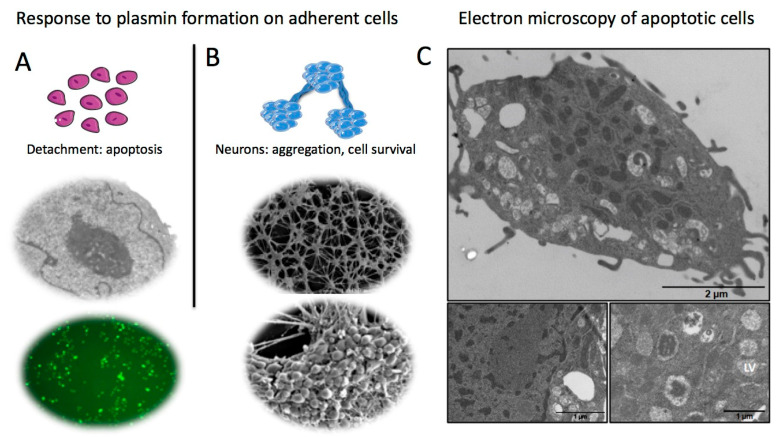
Consequences of plasmin formation on adherent cells. (**A**) Detachment of CHO-1 cells from the substratum upon degradation of extracellular matrix proteins (fibronectin, laminin) by membrane-bound plasmin. Most detached adherent cells enter apoptosis as indicated by nuclear condensation (electron microscopy) and TUNEL positive cells. (**B**) Plasminogen activation on mouse cortical neurons. Upon detachment from the substratum, neurons form interconnected clusters that resist to apoptosis and survive longer. (**C**) Cell changes typical of apoptosis as visualised by transmission electron microscopy (CHO-1 cells). The main and lower panels show late apoptotic changes (chromatin condensation, compaction and fragmentation of the nucleus, vacuolisation of the cytoplasm and lysis vesicles (LV)) [42,50].

**Figure 4 ijms-24-01571-f004:**
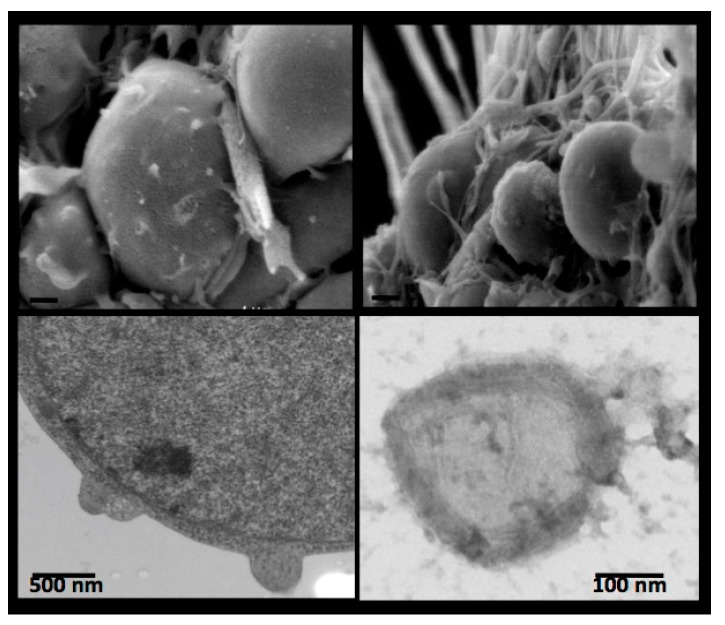
Cell membrane budding and blebbing (mouse cortical neurons) visualised by scanning (**upper** micrographs) and transmission (**bottom** micrographs) electron microscopy. An isolated microvesicle is depicted in the lower right micrograph [42].

**Figure 5 ijms-24-01571-f005:**
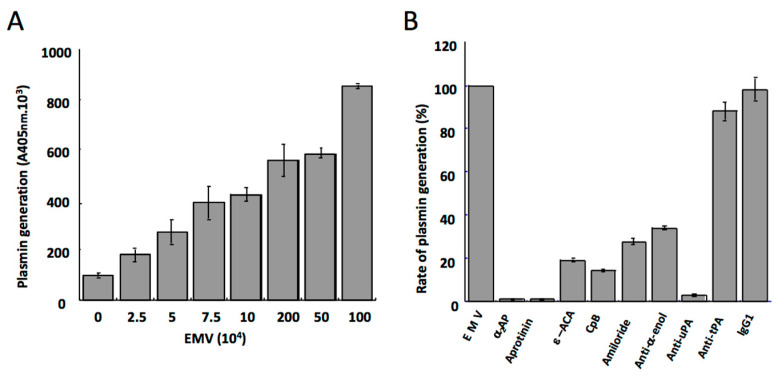
Plasmin generation on membrane microvesicles derived from the human microvascular endothelial cell line HMEC-1. (**A**) Varying concentrations of endothelial microvesicles (EMV) were incubated with a fixed amount of plasminogen. Plasmin generation was detected with a chromogenic substrate. The amount of plasmin formed is a function of the concentration of microvesicles, i.e., of the amount of plasminogen activator (uPA) present at their membrane. (**B**) The specificity and characteristics of the activation of plasminogen at the microvesicle surface. Plasmin formed on the endothelial microvesicles is inhibited by a_2_-antiplasmin (α_2_-AP) and aprotinin. Binding and activation of plasminogen on endothelial microvesicles is prevented by the lysine analogue ε-aminocaproic acid (ε-ACA), carboxypeptidase B (CpB), and an anti-α-enolase (anti-α-enol) polyclonal antibody. The activity of uPA on the microvesicles is inhibited by amiloride and a specific polyclonal antibody anti-uPA, whereas an antibody anti-tPA or non-immune IgG has no effect [29].

**Figure 6 ijms-24-01571-f006:**
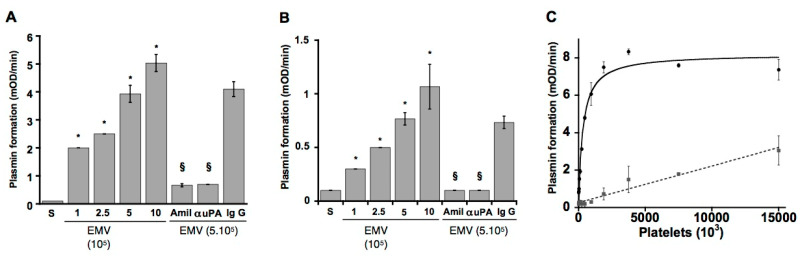
Fibrinolytic cross-talk between endothelial microvesicles (EMV) or cells and fibrin-, fibronectin- or platelet-bound plasminogen. Plasmin is formed when fibrin-bound plasminogen (**A**) or fibronectin-bound plasminogen (**B**) is cleaved by microvesicles carrying uPA. Plasmin formation is prevented by amiloride (Amil) and a polyclonal antibody directed against uPA (α-uPA) as compared to the IgG isotype control. (**C**) Platelet-bound plasminogen is activated by a fixed concentration of uPA-bearing microvesicles in a platelet concentration-dependent manner until saturation. The dotted line represents non-specific activity on platelet-bound plasminogen by tPA-bearing cells (mouse cortical neurons) [95]. * Significant changes compared with activation in supernatants. § significant changes compared with activation on at 5 × 10^5^ EMPs (*p* < 0.05.).

**Figure 7 ijms-24-01571-f007:**
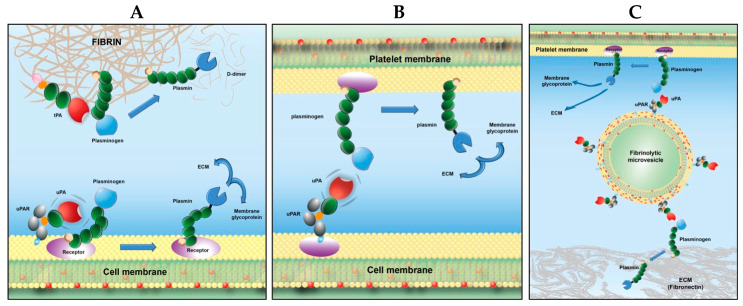
Schematic representation of plasmin formation on biological surfaces [101]. (**A**) Canonical mechanism for plasminogen activation; assemblage of both plasminogen and its activator on the same surface is required for plasmin generation. (**B**) Fibrinolytic cross-talk mechanism on platelets: platelet-bound plasminogen is activated by uPA-bearing cells (e.g., monocytes). (**C**) Fibrinolysis cross-talk mechanism by microvesicles: ECM- or platelet-bound plasminogen is transformed into plasmin by microvesicles bearing uPA.

## Data Availability

Not applicable.

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
