# Peer review of "Distinguishing Plasmin-Generating Microvesicles: Tiny Messengers Involved in Fibrinolysis and Proteolysis"

_ijms, 2023, doi:10.3390/ijms24021571_

Round 1
Reviewer 1 Report
This manuscript provided an overview on the mechanism of plasminogen reception and activation at the surface of cell-derived microvesicles, new actors in fibrinolysis and proteolysis.
A few of suggestions for the authors.
1. Page 2, line 68, “alteration of the of the interactions with the …”?
2. Page 4, line 142 is not showing.
3. Page 4, Figure 2B, [e-ACA] need to be presented with the full name and need to be explained a little bit.
4. Page 7, Figure 5, EMV need to have the full name when presented for the first time.
5. Some of the abbreviations in this manuscript do not have the full name indicated when the authors use them for the first time.
Author Response
Author's Reply to the Review Report (Reviewer 1)
R: English language and style are fine/minor spell check required.
Au: Particular attention was paid to the reviewer's recommendation concerning minor misspelling.
R: Page 2, line 68, “alteration of the of the interactions with the …”?
Au: the phrase has been corrected
R: Page 4, line 142 is not showing.
Au: we repositioned the figure to allow the corresponding line to be visible.
R: Page 4, Figure 2B, [e-ACA] need to be presented with the full name and need to be explained a little bit.
Au: Figure 2B, [e-ACA] is now presented with the full name and further explanation is given.
R: Page 7, Figure 5, EMV need to have the full name when presented for the first time.
Au: EMV in figure 5 has been fully explained.
R: Some of the abbreviations in this manuscript do not have the full name indicated when the authors use them for the first time.
Au: Full name is now indicated for all abbreviations appearing for the first time in the text.

Reviewer 2 Report
A well-written review on the possible role of cell-derived microvesicles in plasminogen activation. There are only some minor points where the manuscript can be improved:
1) Line 142 is missing in the text and should be added
2) Line 105-107: I find this sentence about exosomes rather confusing. the differences and the similarities with the microvesicles in the sentence before are not clearly explained, and nowhere in the rest of the text do the authors mention exosomes again. Therefore i suggest that the authors either delete this sentence or cover the differences between exosomes and microvesicles in more detail.
3) Line 149-150: I suggest rephrasing the first part of this sentence slightly into: "Comparable results were obtained in a variety of cell lines including"
4) line 149-150: The authors should add references for the studies with comparable results
5|) Figure 5: The abbreviation of EMV (X-axis figure 5A) should be explained in the legend. The Greek characters in Ɛ-ACA and anti-α-enol are missing in the X-axis of Figure 5B
6) Figure 6: The abbreviation of EMV (X-axis figure 6A and 6B) should be explained in the legend.
7) There are some problems with the use of singular and plural in sentences: line 30: 'this article are' should be "this article is", line 70: "remodelling lead" should be "remodelling leads" line 138: "blocks" should be "block", line 151: "blocks" should be "block", line 184: "trigger" should be "triggers"
8) other minor linguistic points: Line 68: "of the of the" should be "of the", line 398: "carry" should be "carried"
Author Response
Author's Reply to the Review Report (Reviewer 2)
R: English language and style are fine/minor spell check required.
Au: Particular attention has been paid to the reviewer's recommendation concerning minor spelling mistakes that he observed.
R: 1) Line 142 is missing in the text and should be added
Au: we verified that corresponding line 142 is now showing
2) Line 105-107: I find this sentence about exosomes rather confusing. the differences and the similarities with the microvesicles in the sentence before are not clearly explained, and nowhere in the rest of the text do the authors mention exosomes again. Therefore i suggest that the authors either delete this sentence or cover the differences between exosomes and microvesicles in more detail.
Au: We agree with the Reviewer on the inadequacy of this sentence in this paragraph. In order to avoid confusion, the suggestion of the Reviewer to delete this sentence is welcome.
R: 3) Line 149-150: I suggest rephrasing the first part of this sentence slightly into: "Comparable results were obtained in a variety of cell lines including"
Au: we have rephrased the sentence lines 149-150 as suggested by the Reviewer.
R: 4) line 149-150: The authors should add references for the studies with comparable results
Au: References were added as suggested by the Reviewer
R: 5|) Figure 5: The abbreviation of EMV (X-axis figure 5A) should be explained in the legend. The Greek characters in Ɛ-ACA and anti-α-enol are missing in the X-axis of Figure 5B
Au: The abbreviation of EMV (X-axis figure 5A) is now explained in the legend as requested by the Reviewer. The figure has been revised to introduce the Greek characters missing in the X-axis of Figure 5B.
R: 6) Figure 6: The abbreviation of EMV (X-axis figure 6A and 6B) should be explained in the legend.
Au: The abbreviation of EMV (X-axis figure 6A and 6B) is now explained in the legend, as requested by the Reviewer.
R: 7) There are some problems with the use of singular and plural in sentences: line 30: 'this article are' should be "this article is", line 70: "remodelling lead" should be "remodelling leads" line 138: "blocks" should be "block", line 151: "blocks" should be "block", line 184: "trigger" should be "triggers"
Au: Singular and plural forms of verbs as well as tense accordance have been corrected and revised in the whole manuscript as suggested by the Reviewer.
R: 8) other minor linguistic points: Line 68: "of the of the" should be "of the", line 398: "carry" should be "carried"
Au: We have greatly appreciated the careful reading of our manuscript by the Reviewer and have introduced all recommendations indicated. Careful reading of the manuscript by all authors was made following the specific recommendations of the Reviewer.
